# Mathematical modeling of N-803 treatment in SIV-infected non-human primates

**Jonathan W. Cody**[1], **Amy L. Ellis-Connell**[2], **Shelby L. O'Connor**[2], **Elsje Pienaar**[1]*

**1** Weldon School of Biomedical Engineering, Purdue University, West Lafayette, Indiana, United States of America, **2** Department of Pathology and Laboratory Medicine, University of Wisconsin-Madison, Madison, Wisconsin, United States of America

* epienaar@purdue.edu

**Data Availability Statement:** All relevant data are within the manuscript and its Supporting Information files.

**Funding:** This work was supported by seed funding from the Indiana Center for AIDS Research

## Abstract

Immunomodulatory drugs could contribute to a functional cure for Human Immunodeficiency Virus (HIV). Interleukin-15 (IL-15) promotes expansion and activation of CD8+ T cell and natural killer (NK) cell populations. In one study, an IL-15 superagonist, N-803, suppressed Simian Immunodeficiency Virus (SIV) in non-human primates (NHPs) who had received prior SIV vaccination. However, viral suppression attenuated with continued N-803 treatment, partially returning after long treatment interruption. While there is evidence of concurrent drug tolerance, immune regulation, and viral escape, the relative contributions of these mechanisms to the observed viral dynamics have not been quantified. Here, we utilize mathematical models of N-803 treatment in SIV-infected macaques to estimate contributions of these three key mechanisms to treatment outcomes: 1) drug tolerance, 2) immune regulation, and 3) viral escape. We calibrated our model to viral and lymphocyte responses from the above-mentioned NHP study. Our models track CD8+ T cell and NK cell populations with N-803-dependent proliferation and activation, as well as viral dynamics in response to these immune cell populations. We compared mathematical models with different combinations of the three key mechanisms based on Akaike Information Criterion and important qualitative features of the NHP data. Two minimal models were capable of reproducing the observed SIV response to N-803. In both models, immune regulation strongly reduced cytotoxic cell activation to enable viral rebound. Either long-term drug tolerance or viral escape (or some combination thereof) could account for changes to viral dynamics across long breaks in N-803 treatment. Theoretical explorations with the models showed that less-frequent N-803 dosing and concurrent immune regulation blockade (e.g. PD-L1 inhibition) may improve N-803 efficacy. However, N-803 may need to be combined with other immune therapies to countermand viral escape from the CD8+ T cell response. Our mechanistic model will inform such therapy design and guide future studies.

## Author summary

Immune therapy may be a critical component in the functional cure for Human Immunodeficiency Virus (HIV). N-803 is an immunotherapeutic drug that activates antigen-

[www.niaid.nih.gov/research/centers-aids-research] (to EP) and by National Institutes of Health [www.nih.gov] award number R01AI108415 (to SO). The funders had no role in study design, data collection and analysis, decision to publish, or preparation of the manuscript.

**Competing interests:** The authors have declared that no competing interests exist.

specific CD8$^+$ T cells of the immune system. These CD8$^+$ T cells eliminate HIV-infected cells in order to limit the spread of infection in the body. In one study, N-803 reduced plasma viremia in macaques that were infected with Simian Immunodeficiency Virus, an analog of HIV. Here, we used mathematical models to analyze the data from this study to better understand the effects of N-803 therapy on the immune system. Our models indicated that inhibitory signals may be reversing the stimulatory effect of N-803. Results also suggested the possibilities that tolerance to N-803 could build up within the CD8$^+$ T cells themselves and that the treatment may be selecting for virus strains that are not targeted by CD8$^+$ T cells. Our models predict that N-803 therapy may be made more effective if the time between doses is increased or if inhibitory signals are blocked by an additional drug. Also, N-803 may need to be combined with other immune therapies to target virus that would otherwise evade CD8$^+$ T cells.

## Introduction

In 2019, there was an estimated 38.0 million people living with Human Immunodeficiency Virus (HIV) and 690,000 deaths related to Acquired Immune Deficiency Syndrome (AIDS) [1]. Current antiretroviral therapy (ART) remains a life-long therapy, since treatment interruption inevitably leads to viral rebound [2]. Alternative treatment strategies include reversing latent infections [3], introducing cellular and humoral vaccines [4], enhancing T cell function [5], and enhancing NK cell function [6]. These immune-based approaches could reduce the reliance on continuous and lifelong ART and contribute to a functional HIV cure.

One immunotherapeutic approach involves interleukin-15 (IL-15). Interleukin-15 is a cytokine that induces proliferation and activation of CD8$^+$ T cells and natural killer (NK) cells (reviewed in [7,8]). Although treatment with monomeric IL-15 did not lower plasma viral load in non-human primates (NHPs) infected with Simian Immunodeficiency Virus (SIV) [9,10], treatment with the heterodimeric IL-15/IL-15Rα complex did reduce viral load in plasma and lymph tissue of NHPs infected with Simian/Human Immunodeficiency Virus (SHIV) [11]. N-803 [ImmunityBio] (formerly ALT-803 [Altor Biosciences]) is an IL-15 superagonist that combines an IL-15 variant with improved bioactivity [12] with an IL-15Rα-Fc complex to extend serum half-life and bioavailability [13]. This superagonist induced proliferation of CD8$^+$ T cells and NK cells in healthy NHPs [14,15], SIV-infected NHPs [15,16], and in humans participating in cancer trials [17–19]. In one NHP study, N-803 treatment reduced the number of SIV-infected cells in B-cell follicles but did not consistently lower plasma viral load [15]. In a different cohort of NHPs genetically predisposed to SIV control and vaccinated prior to infection, weekly doses of N-803 successfully lowered SIV viral load in the plasma, though the effect was transient [16]. After initially being suppressed, the viral load partially rebounded during the first month of weekly doses. However, after a 29-week break in treatment, N-803 regained partial efficacy in reducing plasma viral load. Thus, there were variations in treatment efficacy along both short (weeks) and long (months) timescales. While this is only one study, these dynamic responses provide a unique opportunity to quantify transient treatment responses and suggest that changes in treatment scheduling of N-803 could improve efficacy in reducing SIV viral load. However, such optimization would require an understanding of the underlying mechanisms driving the observed loss and recovery of treatment efficacy.

The vaccinated NHP study identified several mechanisms which could have compromised the efficacy of N-803 [16]. We broadly consider these mechanisms in three categories (Table 1). The first mechanism, drug tolerance, was evidenced by the decline of IL-15 receptor

**Table 1. Mechanisms considered to compromise N-803 efficacy.**

| | |
|---|---|
| **Drug Tolerance** | Factors which act only to diminish the stimulatory effect of N-803 on CD8$^+$ T cells and NK cells (e.g. downregulation of IL-15 receptors) |
| **Immune Regulation** | Factors which act to inhibit the immune response of CD8+ T cells and NK cells (e.g. upregulation of immune checkpoint molecules) |
| **Viral Escape** | Selection of SIV variants that evade the CD8$^+$ T cell immune response |

expression by CD8$^+$ T cells and NK cells during N-803 treatment, thereby reducing the available targets for N-803. The second mechanism we term immune regulation. Expression of inhibitory markers (CD39 and PD-1) by CD8$^+$ T cells and NK cells increased, as did the presence of regulatory T cells (CD4$^+$CD25$^+$CD39$^+$ phenotype) in the peripheral blood. In other studies, N-803 increased serum levels of the anti-inflammatory cytokine IL-10 in mice [20], and, in a mouse model of cerebral malaria, N-803 induced NK cells to secrete IL-10, which decreased CD8$^+$ T cell activation in the brain [21]. Together, these data indicate that there may be a systemic anti-inflammatory response that could hamper the ability of N-803 to stimulate prolonged anti-viral immune responses. In this work, we broadly group these anti-inflammatory responses under the term immune regulation. Third, the amino acid sequence of targeted CD8$^+$ T cell epitopes was altered during N-803 treatment, which could be consistent with viral escape [16]. As a result, previously generated CD8$^+$ T cells may not recognize circulating viral variants [22–24]. While evidence of all three of these mechanisms exists in the NHP data, the contributions of each mechanism to the loss and recovery of viral suppression under N-803 therapy have not been quantitatively assessed, a task that is difficult to do experimentally.

Computational models are well-suited to quantify and deconvolute the effects of multiple interacting mechanisms in complex systems. Ordinary differential equation (ODE) models have been used to study HIV and its treatment (reviewed in [25,26]). ODE models have investigated the potential of various treatment strategies, including reactivating latent infections [27,28], cytotoxic cell stimulation [27], and cellular vaccines [29]. Modelers have also explored how immune regulation [30,31] and viral escape [29,32] affect cytotoxic cell function and HIV infection.

Here we combine, for the first time, pharmacokinetics and pharmacodynamics of N-803 with an HIV infection model that includes both cytotoxic T-cell and NK cell populations. We also newly combine this model with mechanisms which may lower N-803 efficacy. These mechanisms are: drug tolerance that weakens the N-803 effect in cytotoxic cells; immune regulatory signals that inhibit cytotoxic cell function; and viral escape from cytotoxic cell targeting. We calibrated the model to data from one vaccinated NHP study, specifically to longitudinal viral, CD8$^+$ T cell, and NK cell measurements from the peripheral blood [16]. We applied the model to quantify how drug tolerance, immune regulation, and viral escape may have contributed to the dynamics of SIV viremia during N-803 treatment in this unique set of NHPs. We also predicted how these mechanisms might impact potential improvements to N-803 regimens.

## Methods

### Mathematical model

**Viral infection.** We followed the practice of representing the within-host dynamics of viral infection with a system of ordinary differential equations [25–32]. Eqs (1–4) describe the model of viral infection and immune response in the absence of N-803 treatment. This is a single-compartment model that does not explicitly consider migration between blood, lymph,

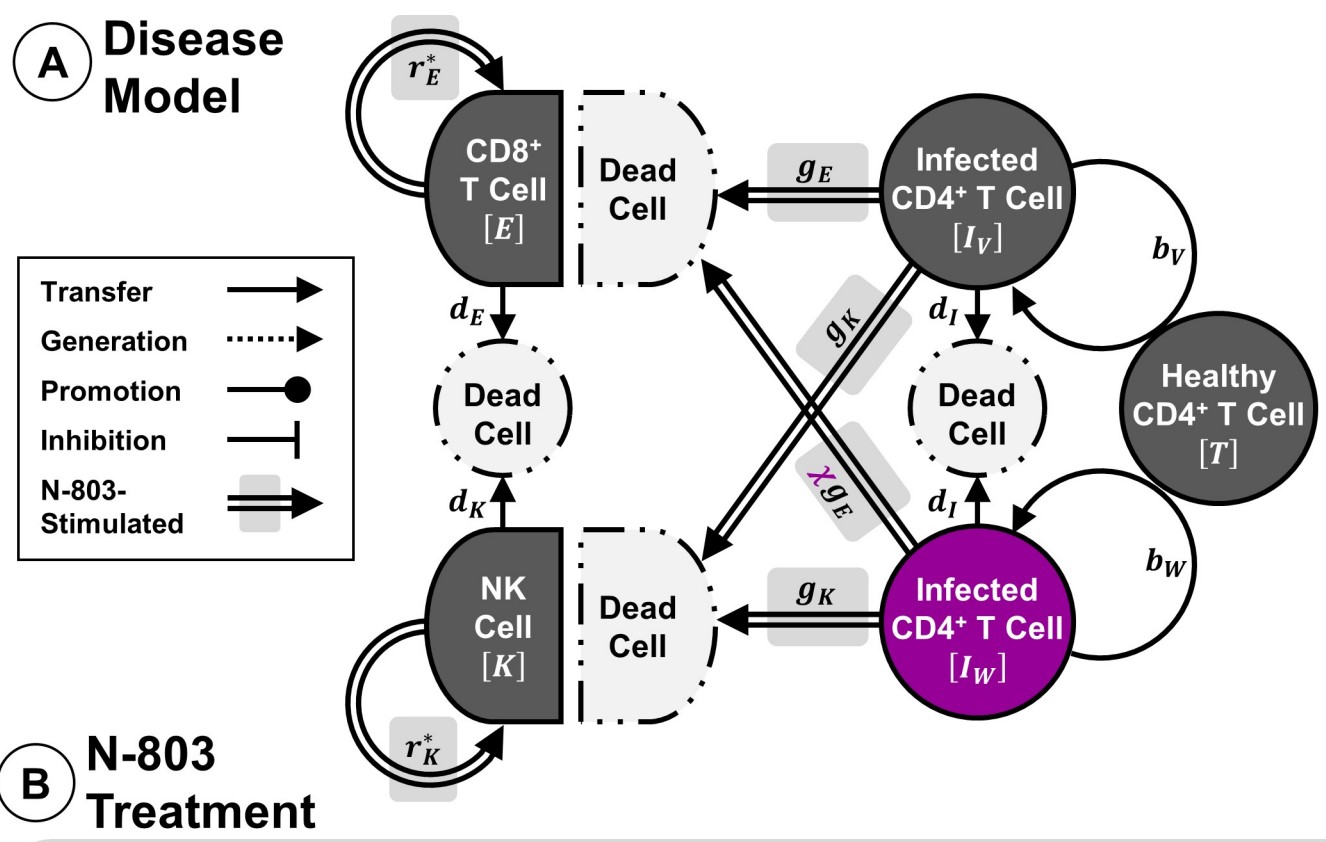

**Fig 1. Mathematical model of N-803 treatment of SIV.** (A) SIV disease model includes cells infected with one of two variants of SIV virus ($I_v$ and $I_w$), along with CD8$^+$ T cells ($E$) and NK cells ($K$) (Eqs 1–4). Proliferation rate constants $r^*_E$ and $r^*_k$ for CD8$^+$ T cells and NK cells are modified by density-dependent terms (not

included in figure, please see Eqs 3 and 4 for details). (B) N-803 treatment model includes pharmacokinetics at absorption site and plasma compartments (Eqs 5 and 6). N-803 stimulates proliferation and cytotoxicity of CD8+ T cells and NK cells, where drug effect is inhibited by tolerance (Eqs 7–12). Immune regulation inhibits proliferation and cytotoxicity of cells (Eqs 7,8,13 and 14). Double lines indicate the sum of drug-induced and constitutive rates.

and peripheral tissues. The disease model is graphically summarized in Fig 1A. Table 2 lists the dependent variables of the model.

The dynamics of cells infected by SIV are represented by Eq (1).

$$I_V' = b_V T I_V - d_I I_V - g_E E I_V - g_K K I_V \tag{1}$$

Infected cells, $I_V$, infect healthy CD4+ T cells, $T$, with rate constant $b_V$. This infection represents both cell-free and cell-to-cell transmission. Infected cells die with rate constant $d_I$. Healthy CD4+ T cells are assumed to be constant, and free virions are assumed to be proportional to infected cells (assumptions are discussed in S1 Appendix). The latter assumption is common in HIV models [29,32–34] and reduces model complexity while still allowing calibration to experimentally measured fold changes in viral load. CD8+ T cells, $E$, and NK cells, $K$, kill infected cells with second-order rate constants $g_E$ and $g_K$, respectively. Killing rate constants ($g_E$ and $g_K$) are applied to the total populations of CD8+ T cells and NK cells (see S1 Appendix). Changes in the frequency of cytotoxically active cells within these two groups are represented by modifications to these average killing rates (see next subsection, 'N-803 treatment'). The latent viral reservoir is an important contributor to viral rebound following ART interruption [35,36]. However, the role of the latent reservoir in the response during and after immunotherapy alone remains unclear [15,37]. Given this uncertainty, the fact that our experimental data indicate relatively short periods of viral suppression [16], and the parameters an explicit viral reservoir would add, this current model does not explicitly account for latent viral reservoir dynamics, following other models of HIV immune therapy [38–40]. When considering N-803 treatment in the context of ART and long-term suppressed viral load [41,42], mathematical models should include a representation of the latent reservoir.

Viral escape from the CD8+ T cell response is a phenomenon documented in both HIV and SIV [22–24]. Our data subjects included two animals with the *Mamu-B*08* MHC class I allele which had received vaccination with *Mamu-B*08* restricted viral epitopes [43]. Sequencing revealed changes in the amino acid composition of *Mamu-B*08 restricted* epitopes after N-803 treatment, changes which could have occurred during viral escape [16]. Viral escape was incorporated into the model by including two viral variants and no mutation between the variants (Eq 2), following Asquith et. al. [44,45].

$$I_W' = b_W T I_W - d_I I_W - \chi g_E E I_W - g_K K I_W \tag{2}$$

**Table 2. Model variables.**

| | Variable | Symbol | Units |
|---|---|---|---|
| **Infection model** | Cell infected with SIV variant $V$ | $I_V$ | #/μL |
| | Cell infected with SIV escape variant $W$ | $I_W$ | #/μL |
| | CD8+ T cells in peripheral blood | $E$ | #/μL |
| | Natural killer cells in peripheral blood | $K$ | #/μL |
| **Treatment model** | N-803 at absorption site | $X$ | pmol/kg |
| | N-803 in plasma | $C$ | pM |
| | Tolerance variables | $TOL_1 \ldots TOL_N$ | - |
| | Regulation variables | $REG_1 \ldots REG_M$ | - |

Dependent variables from Eqs (1–14) and Fig 1, shown with their corresponding symbol and units.

The cells infected with the escape variant, $I_W$, have reduced susceptibility to cytotoxic T cells (by applying a factor $\chi < 1$ to the killing rate). This variant also infects target cells at a lower rate constant ($b_W < b_V$), as escape can often incur a fitness penalty [46–48].

Both CD8+ T cells [49,50] and NK cell [51] populations are maintained by self-renewal (Eqs 3 and 4).

$$E' = r_E \left( \frac{h}{h + E} \right) E - d_E E \tag{3}$$

$$K' = r_K \left( \frac{h}{h + K} \right) K - d_K K \tag{4}$$

CD8+ T cells and NK cells proliferate with rate constants $r_E$ and $r_K$ and undergo apoptosis with rate constants $d_E$ and $d_K$, respectively. To maintain a stable population, proliferation and survival are thought to be density-dependent, which could arise from competition for space and cytokines [52]. Therefore, our proliferation rates are modified by density-dependent terms governed by $h$ [27,53]. Stimulation of CD8+ T cells and NK cells via viral antigen is assumed to remain at a constant or saturated level. Thus, absent N-803 intervention, the immune response is constant, which is a common assumption [25,26].

**N-803 treatment.** The pharmacokinetics for N-803 (Eqs 5 and 6, Fig 1B) follows the basic model for extravascular dosing [54].

$$X' = -k_a X \tag{5}$$

$$C' = k_a \left( \frac{F}{v_d} \right) X - k_e C \tag{6}$$

This describes the quantity of N-803 at the absorption site, $X$, and concentration of N-803 in the plasma, $C$. Parameters $k_a$, $k_e$, $F$, and $v_d$ are the absorption rate constant, elimination rate constant, bioavailability, and volume of distribution, respectively.

N-803 has been demonstrated to expand CD8+ T cells and NK cells in healthy NHPs [14,15] SIV-infected NHPs [15,16], and in humans participating in cancer trials [17–19]. N-803 also increased expression of cytolytic proteins perforin and granzyme B in human CD8+ T cells [14] and NK cells in vitro [55,56] and induced secretion of cytokines IFNγ and TNFα in murine CD8+ T cells and NK cells in vivo [20,57,58]. Therefore, we represented N-803 pharmacodynamics by applying a drug-dependent increase (Eqs 7 and 8, Fig 1B) to both the rates of killing and proliferation for CD8+ T cells and NK cells (parameters $g_E$, $g_K$ and $r_E$, $r_K$ in Eqs 1–4).

$$g_i \rightarrow g_i \left[ 1 + \gamma_i \overbrace{\left( \frac{C}{C_{50} + C} \right)}^{\text{Drug Effect}} \overbrace{\left( \frac{1}{1 + \eta(\text{TOL}_{N-1} + \text{TOL}_N)} \right)}^{\text{Drug Tolerance}} \right] \overbrace{\left( \frac{1}{1 + \lambda \text{REG}_M} \right)}^{\text{Immune Regulation}} \quad i = E, K \tag{7}$$

$$r_i \rightarrow r_i \left[ 1 + \rho_i \left( \frac{C}{C_{50} + C} \right) \left( \frac{1}{1 + \eta(\text{TOL}_{N-1} + \text{TOL}_N)} \right) \right] \left( \frac{1}{1 + \varphi \text{REG}_M} \right) \quad i = E, K \tag{8}$$

Effects saturate for both cell types according to a single parameter, $C_{50}$ (Eqs 7 and 8). The parameters $\gamma_E, \gamma_K$ and $\rho_E, \rho_K$ ('Drug Effect' in Eqs 7 and 8) are the maximum relative increases in killing and proliferation rates, respectively. Parameters $\eta$, $\lambda$, and $\varphi$ determine the strength of

inhibition due to drug tolerance and immune regulation, which are discussed in the following paragraphs.

In the N-803 treated NHPs, expression of the IL-15 receptor subunits, CD122 and CD132, declined in both CD8[+] T cells and NK cells with continued treatment, suggesting a possible tolerance to N-803 [16]. Furthermore, the proliferation of CD8[+] T cells and NK cells was weaker in the second and third treatment cycles compared to the first cycle [16]. We phenomenologically represented drug tolerance by adding a delayed inhibition to the drug effect ('Drug Tolerance' term in Eqs 7 and 8), the timing of which is modeled by Eqs (9–12).

$$\mathrm{TOL_1}' = \delta_{\mathrm{TOL}}\left(\frac{C}{C_{50}+C} - \mathrm{TOL_1}\right) \tag{9}$$

$$\mathrm{TOL}_n' = \delta_{\mathrm{TOL}}(\mathrm{TOL}_{n-1} - \mathrm{TOL}_n) \ \ n = 2, 3, \ldots N-2 \tag{10}$$

$$\mathrm{TOL}_{N-1}' = \delta_{\mathrm{TOL}}(\mathrm{TOL}_{N-2} - (1+\tau)\mathrm{TOL}_{N-1}) \tag{11}$$

$$\mathrm{TOL}_N' = \delta_{\mathrm{TOL}}(\tau\mathrm{TOL}_{N-1} - \tau\mathrm{TOL}_N) \tag{12}$$

The build-up and decay of tolerance is governed by two parameters, $\delta_{\mathrm{TOL}}$ and $\tau$. The additional parameter $\tau$ allows a portion of the drug tolerance to persist long-term and attenuate N-803 stimulation in the third cycle.

N-803 treatment of NHPs also coincided with increases in regulatory T cell counts (CD4[+]CD25[+]CD39[+] phenotype) in the peripheral blood and increases in expression of inhibitory markers CD39 in CD8[+] T cells and PD-1 in NK cells [16]. Other studies found that N-803 increased levels of IL-10 in mice, which decreased cytotoxic T cell activation [20,21], though IL-10 was not collected along with the NHP data used in our study. Taken together, this implicates a number of regulatory mechanisms that could counteract the immune stimulatory impact of N-803. As with drug tolerance, we employed a single phenomenological representation of the effects of these immune regulatory pathways (i.e. regulatory T cells, IL-10, etc.), rather than mechanistically modeling each specific pathway (Eqs 13 and 14, Fig 1B).

$$\mathrm{REG_1}' = \delta_{\mathrm{REG}}\left(\frac{C}{C_{50}+C} - \mathrm{REG_1}\right) \tag{13}$$

$$\mathrm{REG}_m' = \delta_{\mathrm{REG}}(\mathrm{REG}_{m-1} - \mathrm{REG}_m) \ \ m = 2, 3, \ldots M \tag{14}$$

Like tolerance, the timing of the regulatory effect was modeled as a delay from the drug effect, this time according to a single parameter $\delta_{\mathrm{REG}}$. Unlike long-term tolerance, incorporating long-term regulation did not improve model fit to data (see S1 Appendix). For the sake of simplicity, it is therefore assumed that the regulatory signals do not persist across the long break in treatment. Immune regulation directly inhibits CD8[+] T cell and NK cell killing and proliferation, where the parameters $\lambda$ and $\varphi$ determine the strength of inhibition of killing and proliferation, respectively ('Immune Regulation' in Eqs (7 and 8)). All of the parameters governing drug tolerance and immune regulation were assumed to be shared between CD8[+] T cells and NK cells. This was necessary to improve identifiability of those parameters and simplify analysis.

By assuming an approximately steady-state prior to treatment, some parameters were derived. Specifically, the collection of parameters governing cell infection and death ($\beta_V$, $\beta_W$, $T$, $d_I$) were calculated from killing parameters ($g_E$, $g_K$, $\chi$), and proliferation rates of cells ($r_E$, $r_K$)

were calculated from cytotoxic cell parameters ($d_E, d_K, h$). The expressions for derived parameter values (Eqs 15–18) include initial conditions $E_0$ and $K_0$.

$$\beta_V T - d_I := q_V = g_E E_0 + g_K K_0 \tag{15}$$

$$\beta_W T - d_I := q_W = \chi g_E E_0 + g_K K_0 \tag{16}$$

$$r_E = d_E \frac{(h + E_0)}{h} \tag{17}$$

$$r_K = d_K \frac{(h + K_0)}{h} \tag{18}$$

## Experimental data

Our mathematical models were calibrated to a single non-human primate study [16]. Three rhesus macaques, chronically infected with SIVmac239 for at least 1.5 years, were given weekly 0.1 mg/kg subcutaneous doses of N-803. The regimen (Fig 2) consisted of three cycles of four treatments each, with a 2-week break between the first and second cycles and a 29-week break between the second and third cycles. Assays to measure plasma viremia (quantified as SIVmac239 gag copy equivalents/mL plasma), as well as CD8$^+$ T cells and NK cells in the peripheral blood, were used as calibration data. We assume SIVmac239 gag copy equivalents in the plasma to be proportional to SIV virions in the peripheral blood. Additional quantities were measured in the peripheral blood, which here served to inform the model. These include CD4$^+$ T cells, regulatory T cells (CD4$^+$CD25$^+$CD39$^+$ phenotype), CD39 expression in CD8$^+$ T cells, and ki-67, PD-1, CD122, and CD132 expression in CD8$^+$ T cells and NK cells. All animals had been vaccinated with SIV epitopes prior to infection and had previously demonstrated transient SIV control as part of a previous study [43].

## Parameter estimation

Maximum likelihood estimation was used to fit model outputs to plasma viral load, CD8$^+$ T cell peripheral blood count, and NK cell peripheral blood count measured in three rhesus macaques chronically infected with SIV and given an N-803 regimen (Fig 2) [16]. To avoid overinterpretation of individual NHP data, we elected to train our model using all three subjects simultaneously. For completeness, the methods described hereafter were repeated for each individual subject, and the results are included in S1 Appendix.

The error model (Eq 19) assumes independent, identical, and normally distributed error $\varepsilon_i \sim (0, \sigma_i^2)$ for each of three response variables (indexed by $i$), with no error covariance between response variables.

$$\boldsymbol{y}_i = f(\boldsymbol{t}_i, \boldsymbol{\theta}) + \boldsymbol{\varepsilon}_i \ \text{ where } \begin{cases} \boldsymbol{y}_1 = \{log_{10}[(\boldsymbol{V} + \boldsymbol{W})_j / (V_0 + W_0)_j]\} \\ \boldsymbol{y}_2 = \{\boldsymbol{E}_j / (E_0)_j)\} \\ \boldsymbol{y}_3 = \{\boldsymbol{K}_j / (K_0)_j)\} \end{cases} \text{ for } j = 1, 2, 3 \tag{19}$$

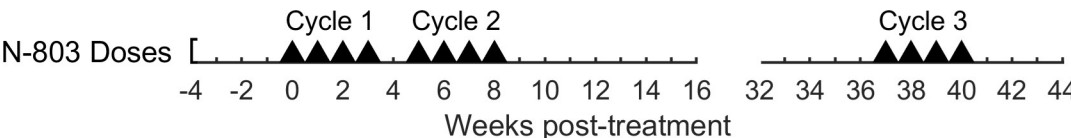

**Fig 2. Dosing schedule for N-803 treatment of SIV-infected NHPs.** Each triangle indicates a 0.1 mg/kg subcutaneous dose of N-803 [16].

Response variables were normalized by initial conditions for each of the three subjects (NHPs, indexed by $j$), as estimated by the mean of the pre-treatment data points. Additionally, virus was log-transformed. Parameter vector $\boldsymbol{\theta}$ was estimated by the concentrated likelihood method [59]. The negative log likelihood (Eq 20) was then a function of the sum of squared error, $S_i$, and the number of data points, $n_i$, for each response variable.

$$\text{NLL}(\boldsymbol{\theta}) = \sum_{i=1}^{3} \frac{n_i}{2}\left[1 + ln\left(\frac{S_i(\boldsymbol{\theta})}{n_i}\right)\right] \tag{20}$$

Some viral data points lay on the lower limit of detection for the viral assay (100 CEQ/mL). We found that either omitting or including these data points did not substantively alter parameter estimation. Therefore, these points were omitted in order to maintain statistical correctness with the likelihood function. Initial parameter estimates were obtained via a multi-start local search approach implemented in MATLAB version R2018b (Mathworks). Further details on parameter estimation can be found in S1 Appendix.

A subset of parameters with sufficient experimental support was fixed during estimation (Table 3) to improve the identifiability of the remaining parameters. For example, pharmacokinetic parameters were fixed at experimental estimates to allow the N-803 50% effective concentration ($C_{50}$) to be identified [14,18]. We used non-human primate data whenever available. Parameters that were not fixed were restricted within biologically feasible ranges, if available (Table 3).

### Uncertainty quantification

In order to quantify the uncertainty of model parameters and predictions, a Bayesian Markov Chain Monte Carlo algorithm was used to sample posterior distributions of the model parameters. Five of the top ten results from the parameter estimation procedure were randomly selected to instantiate a parallel tempering MCMC algorithm [66] that was implemented in the PESTO toolbox [67] for MATLAB. Uniform prior distributions were assumed for all parameters, with boundaries as given in Table 3. The algorithm was run for 400,000 iterations, and the resulting distribution was thinned by selecting every 100th sample. The final sample of 4000 was used for figures and statistical analyses.

### Model comparison

Model comparison was used to identify which model mechanisms (drug tolerance, immune regulation, or viral escape) were required to reproduce the dynamics observed in N-803-treated NHPs. Parameter estimation and uncertainty quantification was performed for the full model (Eqs 1–19) as well as for four additional models (Table 4). Three of the models had either 1) drug tolerance, 2) immune regulation, or 3) viral escape removed. Thus, each of these models combined two of the three mechanisms. The fourth model included only immune regulation (no drug tolerance or viral escape).

Models were compared based on their quantitative and qualitative ability to reproduce the experimental results. The quantitative assessment was done by comparing both the negative log-likelihood (Eq 20) and the Akaike Information Criterion (Eq 21).

$$\text{AIC}_C = 2\text{NLL} + 2\left(n_\theta + n_y\right)\frac{n_t}{n_t - (n_\theta + n_y + 1)} \tag{21}$$

Eq (21) is adapted from the AIC for multivariate regression with small data sets [68]. For our model, the total number of parameters is the length of $\boldsymbol{\theta}$ ($n_\theta$, Table 4), and the number of response variables, $n_y = 3$, since we are neglecting covariance and, thus, have one error

**Table 3. Model parameters.**

| Parameter | Symbol | | Value | Units | Ref. |
|---|---|---|---|---|---|
| Initial SIV virions in plasma [a] | $V_0 + W_0$ | fixed | 3.83 | log(CEQ/ml) | [16] |
| Escape variant initial frequency [a] | $f$ | fitted | (0.001, 1) | | |
| CD8+ T cell killing rate constant | $g_E$ | fitted | $(2 \cdot 10^{-5}, 0.02)$ | μL/#·d | [60] |
| NK cell / CD8+ T cell killing rate ratio [b] | $g_K/g_E$ | fitted | (0.01, 1) | | [61,62] |
| Escape variant susceptibility factor | $\chi$ | fitted | (0.001, 1) | | |
| Initial CD8+ T cells in peripheral blood | $E_0$ | fixed | 520 | #/μL | [16] |
| Initial NK cells in peripheral blood | $K_0$ | fixed | 231 | #/μL | [16] |
| Maximum proliferating cells | $h$ | fitted | (20, 2000) | #/μL | |
| CD8+ T cell death rate constant | $d_E$ | fitted | (0.01, 1) | /day | [63] |
| NK cell death rate constant | $d_K$ | fitted | (0.01, 1) | /day | [63] |
| Initial N-803 at absorption site | $X_0$ | fixed | 880 | pmol/kg | [64] |
| N-803 absorption rate constant | $k_a$ | fixed | 0.80 | /day | [18] |
| N-803 clearance rate constant | $k_e$ | fixed | 2.1 | /day | [14] |
| N-803 vol. of distribution / bioavailability | $v_d/F$ | fixed | 1.3 | L/kg | [14,18] |
| N-803 50% effective concentration | $C_{50}$ | fitted | (1, 1000) | pM | [14,15] |
| CD8+ T cell maximum expansion rate [c] | $\rho_E \cdot d_E$ | fitted | (0.02, 2) | /day | [65] |
| NK cell maximum expansion rate [c] | $\rho_K \cdot d_K$ | fitted | (0.02, 2) | /day | [65] |
| CD8+ T cell killing stimulation factor | $\gamma_E$ | fitted | (0.01, 100) | | |
| NK cell killing stimulation factor | $\gamma_K$ | fitted | (0.01, 100) | | |
| Tolerance effect factor | $\eta$ | fitted | (0.01, 100) | | |
| Proliferation regulation factor | $\varphi$ | fitted | (0.01, 100) | | |
| Killing regulation factor | $\lambda$ | fitted | (0.01, 100) | | |
| Number of tolerance variables | $N$ | fixed | 6 | | [16] |
| Number of regulation variables | $M$ | fixed | 2 | | [16] |
| Tolerance rate constant | $\delta_{\text{TOL}}$ | fitted | (0.05, 5) | /day | |
| Regulation rate constant | $\delta_{\text{REG}}$ | fitted | (0.05, 5) | /day | |
| Tolerance recovery | $\tau$ | fitted | (0.001, 1) | | |

Parameters were either fixed at values shown or restricted within ranges shown during all analysis. Parameters not included in the table were calculated by assuming an approximately steady-state prior to treatment (Eqs 15–18). See S1 Appendix for discussion of values and ranges informed by literature. Ranges for parameters with no measurable experimental analog were intentionally broad.

[a] We assume SIVmac239 gag copy equivalents in the plasma to be proportional to SIV virions in the peripheral blood. The initial conditions for the virus variants $V$ and $W$ were determined from the total initial viral load ($V_0 + W_0$) and the initial frequency of variant $W$ ($f$).

[b] The value of the NK cell killing rate constant $g_K$ is defined as some fraction of CD8+ T cell killing rate constant $g_E$.

[c] The value of proliferation stimulation factors $\rho_E$, $\rho_K$ are defined by the maximum expansion rates of their respective populations.

parameter for each response variable. The parameter penalty term is further modified by number of data points. Since each response variable had a different number of data points, we take their average, $n_t = 280/3$.

**Table 4. Summary of models compared.**

| | Drug Tolerance | Immune Regulation | Viral Escape | Parameters Removed | Parameter Count ($n_\theta$) |
|---|---|---|---|---|---|
| Control | √ | √ | √ | none | 27 |
| Model #1 | | √ | √ | $N, \delta_{\text{TOL}}, \tau, \eta$ | 23 |
| Model #2 | √ | | √ | $M, \delta_{\text{REG}}, \varphi, \lambda$ | 23 |
| Model #3 | √ | √ | | $f, \chi$ | 25 |
| Model #4 | | √ | | $N, \delta_{\text{TOL}}, \tau, \eta, f, \chi$ | 21 |

Variant models #1-#4 were created by fixing select parameters at zero in Eqs (1–18). The parameter count ($n_\theta$ in Eq 21) is the remaining number of fixed and fitted parameters (Table 3).

Three quality metrics were formulated based on the observed viral response to each cycle of treatment [16]. This assessment focused on the viral load, as it is the most relevant treatment outcome. The metrics are mathematically defined in S1 Appendix. Briefly, the log fold rebound in cycle 1 is the difference between the minimum virus in cycle one and the virus at the end of treatment cycle 1 (week 4). The two remaining criteria quantify the observation that viral suppression was largest in cycle 1, followed by that of cycle 3, then cycle 2. Thus, these metrics compare the log fold drops of these cycles (difference between virus at the start of the cycle and the minimum virus during that cycle).

## Per-cell killing (PCK) and related equations

In order to quantify the effect of immune regulation, drug tolerance, and viral escape on per-cell cytotoxic activity, we defined per-cell killing (PCK, Eqs 22–24). The PCK is mathematically equivalent to the second order rate parameter for cytotoxic action if it were applied to the total infected cell population ($I_V + I_W$) and total cytotoxic cell population ($E + K$).

$$\text{PCK} = \frac{[\text{total killing rate}]}{(E+K)(I_V+I_W)} = \frac{([v]+\chi[w])g_E(1+\gamma_E[\Theta][\Omega])[e]+g_K(1+\gamma_K[\Theta][\Omega])[k]}{1+\lambda[\text{REG}_M]} \quad (22)$$

$$[\Theta] = \left(\frac{[C]}{C_{50}+[C]}\right) \quad (23)$$

$$[\Omega] = \left(\frac{1}{1+\eta([\text{TOL}_{N-1}]+[\text{TOL}_N])}\right) \quad (24)$$

The effects of immune regulation, drug tolerance, and viral escape on PCK were calculated as follows (Eqs 25–27). The fold change in PCK due to one mechanism was quantified as the ratio of per-cell killing (PCK) with that mechanism to the PCK without that mechanism.

Immune regulation:

$$\frac{\text{PCK}}{\text{PCK}(\lambda=0)} = \frac{1}{1+\lambda[\text{REG}_M]} \quad (25)$$

Drug tolerance:

$$\frac{\text{PCK}}{\text{PCK}(\eta=0)} = \frac{([v]+\chi[w])g_E(1+\gamma_E[\Theta][\Omega])[e]+g_K(1+\gamma_K[\Theta][\Omega])[k]}{([v]+\chi[w])g_E(1+\gamma_E[\Theta])[e]+g_K(1+\gamma_K[\Theta])[k]} \quad (26)$$

Viral escape:

$$\frac{\text{PCK}}{\text{PCK}(\chi=1)} = \frac{([v]+\chi[w])g_E(1+\gamma_E[\Theta][\Omega])[e]+g_K(1+\gamma_K[\Theta][\Omega])[k]}{g_E(1+\gamma_E[\Theta][\Omega])[e]+g_K(1+\gamma_K[\Theta][\Omega])[k]} \quad (27)$$

We introduced a measure of viral fitness by calculating the fold change in the overall viral proliferation rate during treatment (Eq 28), where $q_V$ and $q_W$ are collections of constants (Eqs 15 and 16).

$$\begin{bmatrix} \text{fold change in} \\ \text{viral fitness} \end{bmatrix} \propto \begin{bmatrix} \text{fold change in} \\ \text{viral proliferation} \end{bmatrix} = \frac{q_V[v]+q_W[w]}{q_V(1-f)+q_w f} \quad (28)$$

Fold change in CD8$^+$ T cell and NK cell proliferation due to immune regulation and drug tolerance (Eqs 29 and 30) were calculated in a manner similar to fold changes in per-cell killing due to immune regulation and drug tolerance. For example, the fold change in CD8$^+$ T cell

proliferation due to immune regulation is defined by the ratio of the proliferation rate with immune regulation over the proliferation rate without immune regulation ($\varphi = 0$).

Immune regulation:

$$\frac{\text{Proliferation}}{\text{Proliferation}(\varphi = 0)} = \frac{1}{1 + \varphi[\text{REG}_M]} \tag{29}$$

Drug tolerance:

$$\frac{\text{Proliferation}}{\text{Proliferation}(\eta = 0)} = \frac{1 + \rho_i[\Theta][\Omega]}{1 + \rho_i[\Theta]} \quad i = E, K \tag{30}$$

The numerical data used to generate figures is included in the spreadsheet S1 Data.

## Results

### Mathematical model reproduced key aspects of observed dynamics of SIV viremia, CD8$^+$ T cells, and NK cells during N-803 treatment

The full model (Eqs 1–18) was fitted to SIV in the plasma, and CD8$^+$ T cells and NK cells in the peripheral blood, and Bayesian 95% credible intervals were obtained by MCMC sampling (Fig 3). The data used to create all subsequent figures is available in S2 Data. The model reproduced four key characteristics of the SIV plasma viral load during N-803 treatment (Fig 3). First, the viral load fell sharply during the first 1-2 weeks of treatment (1.00-1.33 log reduction in the model; 1.43-2.08 log reduction in the NHP data) and began to rebound between the first and third week. Second, treatment cycle 2 had a much smaller effect as compared to treatment cycle 1. Third, after treatment cycle 2, the viral load settled to a lower set-point (0.52-0.74 log below pre-treatment viral load in the model; 0.38-0.76 log below in the NHP data). Fourth, the viral response to treatment cycle 3 was similar to the response in treatment cycle 1 but less pronounced (0.13-0.51 log reduction in the model; 0.67-1.30 log reduction in the NHP data).

The model reproduced two characteristics of the response of peripheral blood CD8$^+$ T cells and NK cells to N-803 (Fig 3B and 3C). First, CD8$^+$ T cells rose quickly in the first week (2.4- to 4.7-fold in the model; 2- to 4-fold in the NHP data), and NK cells expanded even further (4.6- to 9.1-fold in the model; 1.5- to 10.5-fold in the NHP data). Second, both cell populations began to contract in the blood after 1 week of treatment. Although the model attributes this contraction to cell death, it may have also been due to cell migration out of the blood. IL-15 has been shown to promote migration to lymph tissue [15,69].

### Immune regulation, coupled with either drug tolerance or viral escape, can reproduce the viral trajectory

The full model of N-803 treatment of SIV (Fig 1) includes three broad mechanisms that can contribute to reduced N-803 efficacy over time: drug tolerance, immune regulation and viral escape. Drug tolerance represents factors which reduce the cells susceptibility to N-803 long-term (Eqs 7–12), such as the downregulation of surface receptors. Immune regulation represents mechanisms that directly inhibit CD8$^+$ T cell and NK cell proliferation and activation short-term (Eqs 7,8,13 and 14), which may include increased expression of PD-1 and CD39, increased presence of regulatory T cells, or increased presence of IL-10. Viral escape represents selection of SIV variants that are not recognized by existing CD8$^+$ T cells (Eqs 1 and 2). The importance of drug tolerance, immune regulation, and viral escape to the dynamics of SIV during N-803 treatment regimen was assessed by systematically removing each mechanism and recalibrating the model, comparing to the full model as a control (Fig 4). Models were

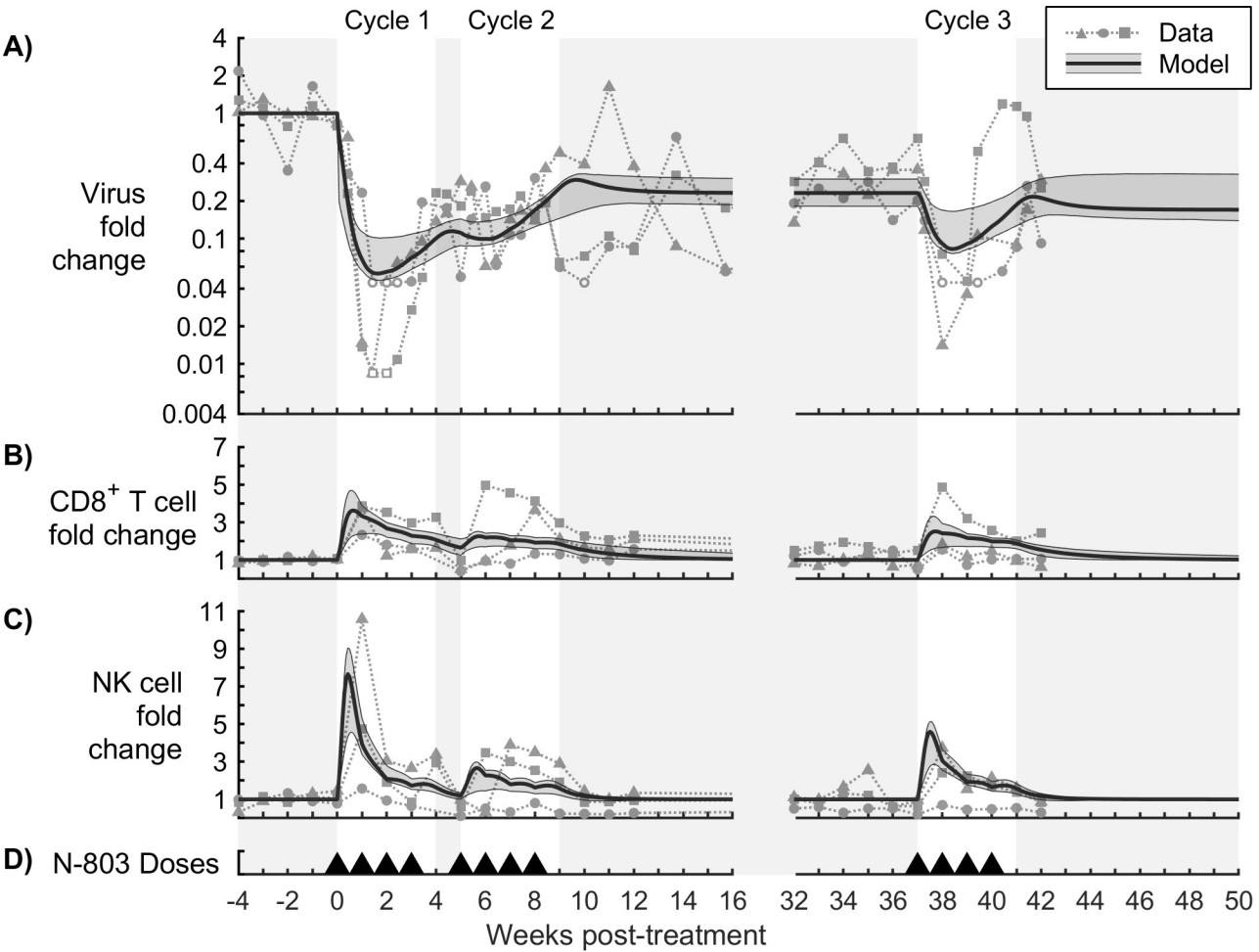

**Fig 3. Model calibration to N-803-treated SIV-infected NHP data.** The model was calibrated to (A) fold change in virus in the plasma, (B) fold change in CD8+ T cells in the peripheral blood, and (C) fold change in NK cells in the peripheral blood. The bold line corresponds to the best-fit model, and the shaded region corresponds to the Bayesian 95% credible interval. See S1 Fig for corresponding parameter distributions. Data from N-803-treated SIV-infected NHPs are shown as different symbols for each NHP [16]. Open symbols were at the lower limit of detection for the viral assay (100 CEQ/mL) and were omitted from parameter estimation. Panel (D) shows timing of 0.1 mg/kg subcutaneous doses of N-803.

compared quantitatively using Negative Log-Likelihood (NLL, Eq 20, Fig 4G), a measure of model fit to the data, and Akaike Information Criterion (AICc, Eq 21, Fig 4H), which also considers model simplicity. We also considered key characteristics of the viral data that should be present in a suitable model (Fig 4F). First, there was a viral rebound in treatment cycle 1 (Fig 4I). Second, the viral response in cycle 2 was weaker than that in cycle 3 (Fig 4J). Third, the response in cycle 3 was weaker than that in cycle 1 (Fig 4K). These collectively represent changes in the short-term and long-term response that should be present in the model.

Without immune regulation (model #2, Fig 4B), the model failed to meet all the quality criteria. Specifically, the virus only decayed to a post-treatment set point, instead of rebounding during the first treatment cycle (Fig 4I). This demonstrates that immune regulation is required in order to represent the short-term (within treatment cycle) viral rebound dynamics. Furthermore, the viral response in cycle 3 was largely lost, showing little improvement in efficacy with respect to cycle 2 (Fig 4J). The model with immune regulation alone (model #4, Fig 4D) also could not adequately replicate the data, showing only a small rebound in cycle 1 (Fig 4I) and

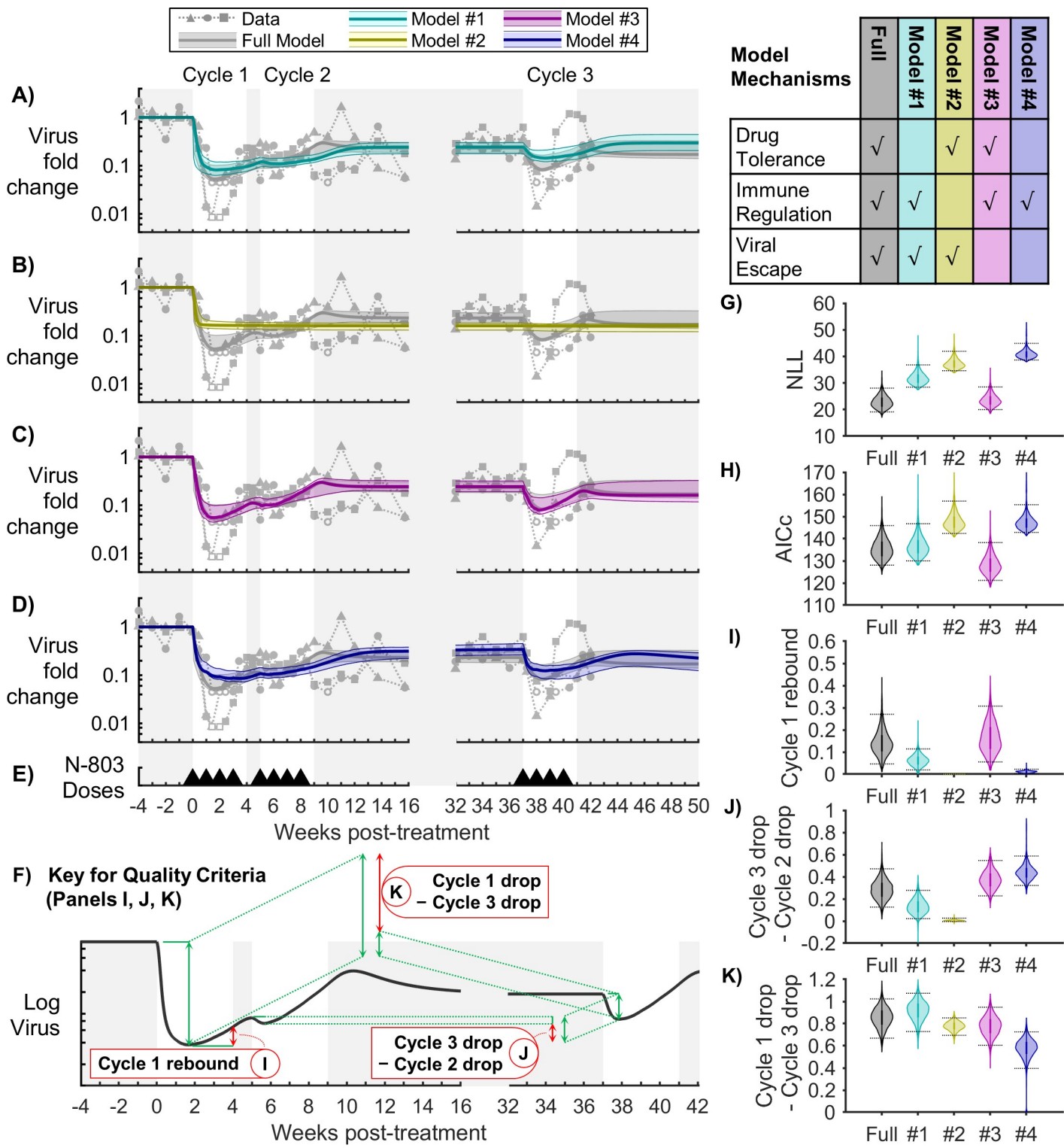

**Fig 4. Model comparison for viral load.** Models with different combinations of mechanisms were compared to assess the importance of drug tolerance, immune regulation, and viral escape. Panels (A-D) compare the fold change in virus between the full model and models #1-4, respectively. The bold line corresponds to the best-fit model, and the shaded region corresponds to the Bayesian 95% credible interval. Data from N-803-treated SIV-infected NHPs are shown as different symbols for each NHP [16]. Open symbols were at the lower limit of detection for the viral assay (100 CEQ/mL) and were omitted from parameter estimation. Panel (E) shows timing of 0.1 mg/kg subcutaneous doses of N-803. Panels (G,H) show the corresponding Negative Log-Likelihood (NLL, Eq 20) and Akaike Information Criterion (AICc, Eq 21) for the Bayesian MCMC samples. Panels (I-K) show the three quality criteria, which are described in panel (F). Bayesian 95% credible intervals are marked. Multiple comparison tests on the quality criteria (I-K) showed a statistically significant difference between all models (p<0.01).

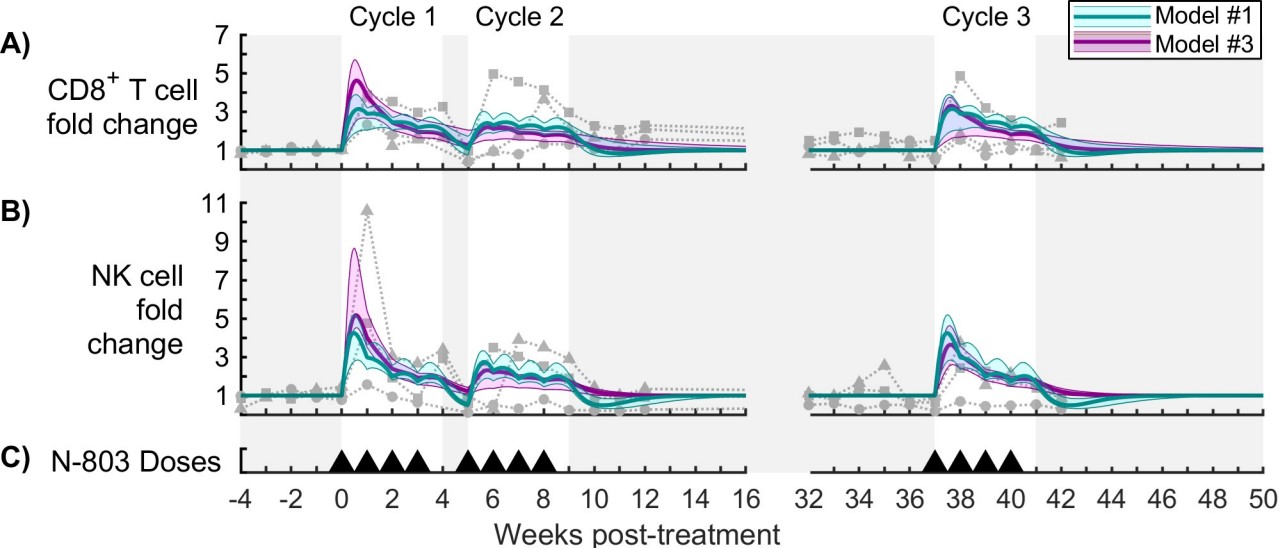

**Fig 5. Model comparison for cytotoxic cells.** Panels (A,B) show fold change in CD8⁺ T cells and NK cells in the peripheral blood, respectively, for the model without drug tolerance (cyan model #1) and the model without viral escape (magenta model #3). The bold line corresponds to the best-fit model, and the shaded region corresponds to the Bayesian 95% credible interval. See S1 Fig for corresponding parameter distributions. Data from N-803-treated SIV-infected NHPs are shown as different symbols for each NHP [16]. Panel (C) shows timing of 0.1 mg/kg subcutaneous doses of N-803. See S2 Fig for models #2 and #4.

the smallest difference between cycles 1 and 3 (Fig 4K). This model also required a depression NK cell counts following cycle 1 (S2 Fig), which was a response observed in only one of the three subjects. Models #2 and #4 also had the highest NLL and AICc scores (Fig 4G and 4H), reflecting poorer agreement with the data. This suggest that immune regulation alone cannot replicate both the short- and long-term responses.

The model without drug tolerance (model #1, Fig 4A) and the model without viral escape (model #3, Fig 4C) both reproduced key characteristics of the viral load (Fig 4I, 4J and 4K). The viral trajectories of these two models were comparable to the full model. Taken together with the results for models #2 and #4, this implies that either drug tolerance or viral escape could have accounted for the long-term changes in viral response. Both models #1 and #3 had comparable or better AICc with respect to the full model (Fig 4H), with model #3 being quantitatively the best model. The higher NLL and AIC for the model without drug tolerance (model #1 compared to model #3 without viral escape) was due in part to a poorer fit to the CD8⁺ T cell and NK cell dynamics (Fig 5). When fitting to individual subjects, these model comparison results held for two out of three NHPs, with the third being inconclusive (possibly due censoring, see S1 Appendix).

### Model quantifies substantial loss in per-cell cytotoxic activity during the course of N-803 treatment

We used the two minimal models (model #1 with immune regulation and viral escape; model #3 with immune regulation and drug tolerance) to quantify the timing and strengths of drug tolerance, immune regulation, and viral escape required to reproduce the observed viral dynamics during N-803 treatment. To this end, we defined a per-cell killing (PCK) metric that can be calculated from fitted parameter values (Eqs 22–24). The PCK is mathematically equivalent to the average rate of killing per infected cell per cytotoxic cell. In other words, multiplying the PCK by the sum of the cytotoxic cells (CD8⁺ T cells, $E$, and NK cells, $K$) and the sum of

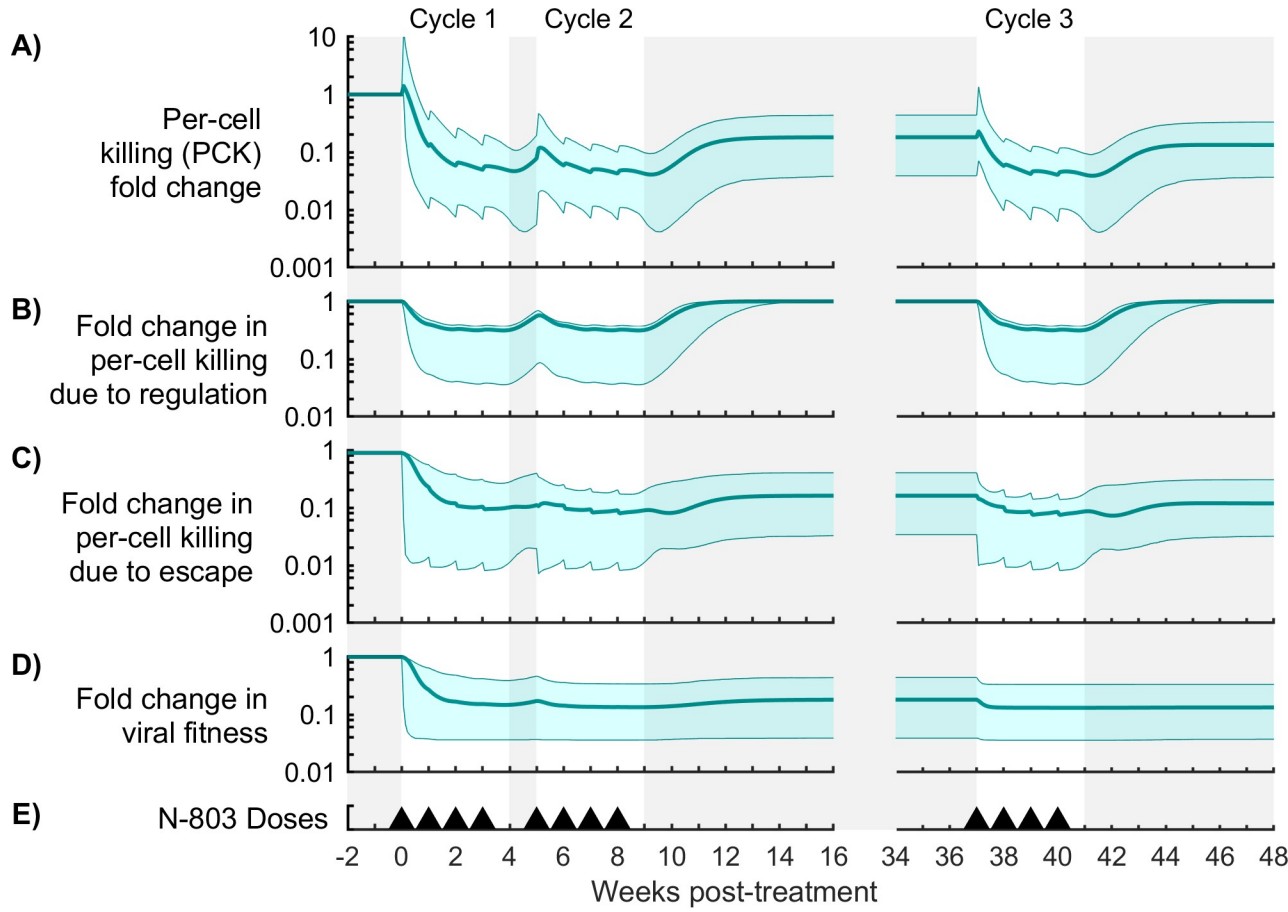

**Fig 6. Contributions of immune regulation and viral escape to per-cell killing (PCK) for Model #1.** Shown are measures of mechanism contribution for the model with immune regulation and viral escape (model #1). Panel (A) shows the fold change in per-cell killing rate, or PCK (Eqs 22–24). Panels (B,C) show the effect of immune regulation and viral escape on PCK (Eqs 25 and 27). Panel (D) shows a measure of viral fitness (Eq 28). The bold line corresponds to the best-fit model, and the shaded region corresponds to the Bayesian 95% credible interval. Panel (E) shows timing of 0.1 mg/kg subcutaneous doses of N-803.

infected cells (both viral variants, $I_V$ and $I_W$) will yield the total rate of loss of infected cells due to cytolytic action. The fold change in PCK due to immune regulation (Eq 25) was quantified by the ratio of per-cell killing with immune regulation to the PCK without immune regulation (PCK($\lambda$=0)). The effect of drug tolerance on PCK (Eq 26) and the effect of viral escape on PCK (Eq 27) were defined similarly.

Despite a brief increase in PCK, both models predict a significant reduction in per cell killing capacity by week 2 (Figs 6A and 7A). Relative to pre-treatment values, PCK fell by 0.76-2.13 log in model #1 and 0.42-0.93 log in model #3 (Bayesian 95% credible intervals), allowing the viral load to rebound within the first treatment cycle while CD8[+] T cells and NK cells were still elevated (Fig 5). Immune regulation caused a 0.41-1.39 log reduction in PCK by week 2 in model #1 (Fig 6B) and a 0.51-1.27 log reduction in model #3 (Fig 7B). Both models predict a recovery in PCK after treatment cycles (week 4 and week 9), which coincided with recovery from immune regulation. Thus, immune regulation both strongly inhibited cytotoxicity during treatment and abated as the cytotoxic cell population normalized after treatment, precluding a post-treatment surge in viremia.

In model #1, viral escape also strongly reduced PCK (0.39-1.87 log reduction by week 2, Fig 6C). However, viral escape was more persistent than immune regulation, maintaining PCK at 0.36-1.41 log below pre-treatment killing rates between cycles 2 and 3 (Fig 6A). This escape

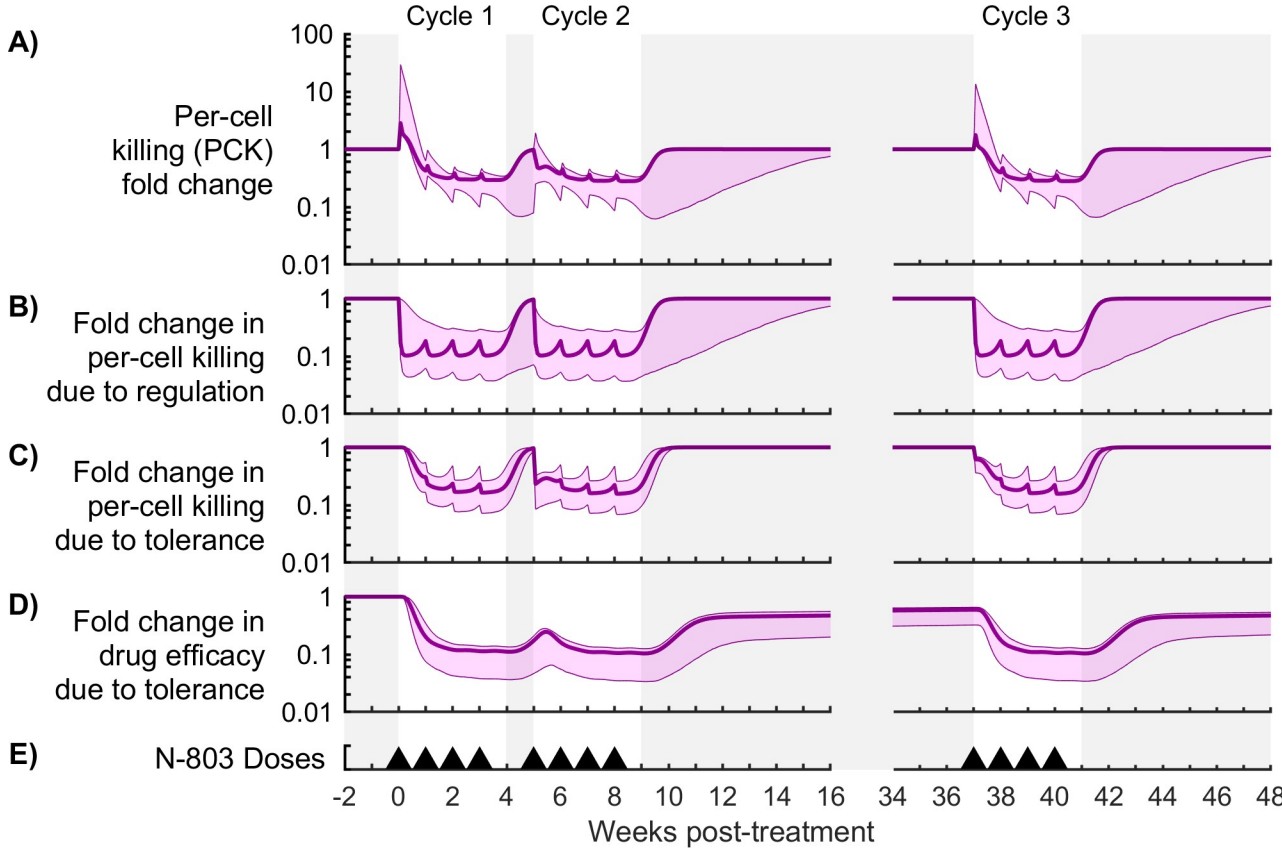

**Fig 7. Contributions of immune regulation and drug tolerance to per-cell killing (PCK) for Model #3.** Shown are measures of mechanism contribution for the model with immune regulation and drug tolerance (model #3). Panel (A) shows the fold change in per-cell killing rate, or PCK (Eqs 22–24). Panels (B,C) show the effect of immune regulation and drug tolerance on PCK (Eqs 25 and 26). Panel (D) shows the effect of tolerance on drug efficacy (Eq 24). The bold line corresponds to the best-fit model, and the shaded region corresponds to the Bayesian 95% credible interval. Panel (E) shows timing of 0.1 mg/kg subcutaneous doses of N-803.

from the CD8[+] T cell response was accompanied by a balancing reduction in viral fitness, which was estimated from the model by the fold change in the total viral proliferation rate (Fig 6D, Eq 28). In model #3, drug tolerance also reduced PCK comparable to immune regulation (0.31-0.96 log reduction by week 2, Fig 6C). Drug tolerance also reduced CD8[+] T cell and NK cell proliferation by 0.22-0.70 log and 0.15-0.59 log, respectively (S3 Fig, Eq 29), while immune regulation had a negligible effect on proliferation in this model. A fraction of drug tolerance persisted across the treatment gap between cycles 2 and 3 (Fig 7D), resulting in the first dose of cycle 3 (week 37) being 34-68% less effective than the first dose of cycle 1. Thus, drug tolerance reduced viral suppression in cycle 3 partly by modulating the proliferative response of CD8[+] T cells and NK cells to N-803. These and other observations were also supported by global sensitivity analysis (see S1 Appendix). The timing of immune regulation and drug tolerance in the model is consistent with expression of inhibitory markers (PD-1 and CD39) and IL-15 receptor subunits (CD122 and CD132) (further details in S1 Appendix).

## N-803 treatment outcome can be improved by larger dosing periods and simultaneous regulatory blockade

To test if the effects of immune regulation, drug tolerance, and viral escape can be overcome through treatment regimen changes, we predicted the impact of dosing periods and

combination therapy on N-803 efficacy. We used both model #1 (immune regulation and viral escape) and model #3 (immune regulation and drug tolerance). Two treatment alternatives were tested: increasing time between doses; and blocking immune regulatory pathways.

Delivering 0.1 mg/kg subcutaneous N-803 doses at 2, 3, and 4 weeks apart yielded lower viral loads in both models, as compared to the current 1-week regimen (Figs 8A, S4A, and S5A). Delivering doses 4 weeks apart resulted in a post-treatment viral load that was 0.09-0.58 log below that of the original regimen for the model #1 and 0.65-3.85 log below for model #3 (Bayesian 95% credible intervals). Dose spacing provided the greatest benefit in model #3 because longer windows between doses allowed more time for immune regulation and drug tolerance to abate (S5B and S5C Fig). In model #1, similar recovery from immune regulation was observed (S4B Fig), but treatment still ultimately resulted in selection of the T cell escape variant (S4C Fig). Taken together this indicates that dosing frequency changes are most likely to improve treatment outcomes if drug tolerance plays a significant part in the observed long-term NHP viral responses.

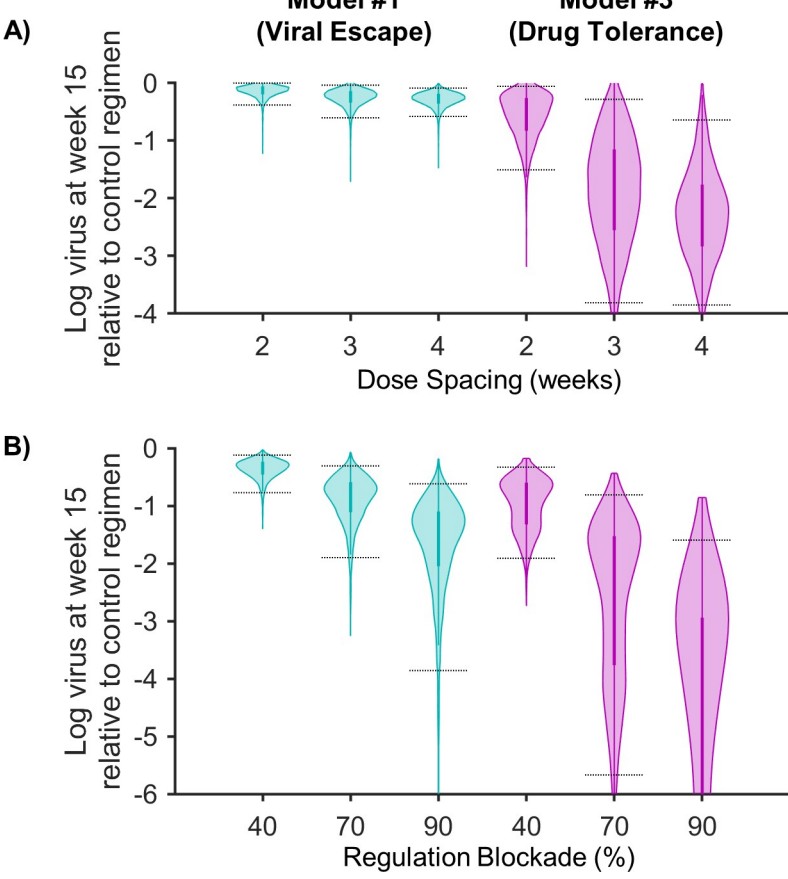

**Fig 8. Potential for regimen changes to improve N-803 treatment outcome.** Shown is a summary of the results of treatment exploration for both the model with immune regulation and viral escape (model #1, left column) and the model with immune regulation and drug tolerance (model #3, right column). Both panels show the log difference in viral load at week 15 as compared to the control regimen (Fig 2 dosing schedule, with no regulation blockade). Panel (A) shows the difference with 2-, 3-, and 4-week dosing regimens. Panel (B) shows the difference with 40, 70 and 90% regulatory blockade (% reductions of killing regulation parameter λ). Bayesian 95% credible intervals are marked. With the exception of changing dose spacing for model #1 (panel A, left), all results were different from zero and different from each other with p<0.01. (See S1 Appendix for statistical method).

The second regimen change we explored was to reduce the killing regulation parameter ($\lambda$) to reflect the potential addition of a drug that blocks regulatory pathways (e.g. PD-1 or PD-L1 antagonists [70–73]). Reducing killing regulation ($\lambda$) by 40, 70, and 90% resulted in lower viral loads in both models (Figs 8B, S4D, and S5D). A 40% reduction in $\lambda$ resulted in a 0.12-0.77 log lower average post-treatment viral load for model #1, compared to the original regimen, and a 0.33-1.91 log lower viral load for model #3. For model #3, the impact of reducing killing regulation was greater for cases when regulation acted early in the treatment cycle (S5D Fig). In model #1, increasing regulatory blockade yielded only small changes in the early viral reduction (S4D Fig) and hastened selection for T cell resistance (S4F Fig). In summary, blockade of immune regulation was consistently effective during weekly N-803 treatment in both models, but, if viral escape is limited (as assumed in model #3), it is especially effective early in the dosing period. Biologically, viral escape could be limited if the CD8+ T cell responses are targeted to conserved viral epitopes [74,75].

## Discussion

We presented novel mathematical models representing immunotherapy of HIV through cytotoxic cell stimulation with an IL-15 superagonist (N-803). We combined the pharmacokinetics and pharmacodynamics of N-803 with an HIV infection model that includes cytotoxic T-cell and NK cell populations as well as experimentally identified mechanisms that lower N-803 efficacy: drug tolerance, immune regulation, and viral escape. The models were applied to analyze data collected from NHPs infected with SIV and treated with three cycles of N-803 [16]. The models reproduced key aspects of the viral and cytotoxic cell trajectories measured in the NHPs, including the transient suppression of viral load with weekly dosing and the partial recovery of drug efficacy following a 29-week break in treatment. Our models predicted how the cytotoxic effector functions of CD8+ T cells and NK cells were diminished during treatment, resulting in rebound of the viral load during treatment. Model comparison suggested that immune regulatory pathways played an important role in the suppression of cytotoxic activity, as this mechanism was required for the model to reproduce viral dynamics in the first treatment cycle. Either drug tolerance or viral escape (or some combination thereof) were capable of accounting for the diminished response of the viral load to the third treatment cycle (relative to the first). The models predicted that adjusting the dosing period of N-803 or complementing with regulatory blockade could improve treatment outcomes. However, the ultimate effectiveness of N-803 monotherapy could be limited by viral escape from the CD8+ T cell response.

We investigated two approaches to countermanding regulatory signals during N-803 treatment. First, we predict that simultaneous blockade of regulatory signals, along with N-803 treatment, could preclude the viral rebound observed during a weekly N-803 regimen, even if viral escape from the CD8+ T cell response is a strong factor. Such combination of PD-1/PD-L1 blockade and an IL-15 agonist has shown promise against cancer in vitro [73]. Furthermore, blockade of the PD-1 pathway via anti-PD-1 or anti-PD-L1 antibodies in the absence of N-803 improved CD8+ T cell function and reduced viral load in SIV-infected NHPs [70,71] and increased HIV-1-specific CD8+ T cell cytotoxicity in some participants in a clinical trial [72]. Our results suggest that IL-15-superagonist and PD-1/PD-L1 blockade combination therapy could be effective against HIV. The second method of countermanding regulation is extending the length of time between N-803 doses. Our model indicates that, when initiating N-803 treatment, there may be a period of cytotoxic stimulation before immune suppression. If subsequent doses are administered after the regulatory signal has abated, stronger efficacy can be achieved for each dose. Rigorous dosing optimization would require a model with

more detailed representation of regulatory pathways such as PD-1, as well as experimental data that frequently measures inhibitory marker dynamics during the critical first week after an N-803 dose. Nonetheless, our model indicates that doses spaced at least 2 weeks apart could improve N-803 efficacy.

N-803 immunotherapy may be perturbing the disease system in ways that persist long term, as evidenced by changes in the proliferative response of NK cells in the third cycle of treatment (drug tolerance) and by changes in the sequences of CD8[+] T cell epitopes in the viral population (viral escape) [16]. Our model demonstrated that long-term changes could be the result of either or both of these phenomena. Further studies will be needed to better quantify the relative contribution of drug tolerance and viral escape in NHPs, and these contributions will affect further N-803 treatment development. If drug tolerance is the main driving mechanism behind the long-term response to N-803 treatment, the effect of tolerance may be circumvented with an optimized dosing regimen. In contrast, while viral escape did not completely preclude a successful N-803 regimen, it limited the impact of all treatment changes (compare model #3 to model #1).

N-803 also has potential to be combined with other therapeutic approaches. There may be a need to couple N-803 with a vaccine that elicits CD8[+] T cell responses targeting conserved viral epitopes. It was postulated that the viral suppression observed in the NHPs used for this study was enabled by the vaccine status of the animals [16]. Additionally, N-803 was shown to enhanced antibody dependent cell-mediated cytotoxicity (ADCC) in NK cells against human cancer cells [76], suggesting a potential for synergy with bnAbs. In ART-treated, SIV-infected NHPs, delivering either therapeutic vaccines [77] or broadly neutralizing antibodies (bnAbs) [78], in combination with an activator (a TLR7 agonist), delayed viral rebound after ART cessation. Future iterations of this model could include ART suppression of viral production [33,34]. Such inclusion would likely require a detailed representation of latent viral infection to account for the importance of the viral reservoir in viral rebound as well as the potential latency reversal effects of N-803 [41,42,79].

Infected cells can evade detection by either not actively producing virions (latent infection [3,80,81]) or by existing in immune privileged tissues (e.g. central nervous system [82] or B-cell follicles in lymph nodes [83]). N-803 has interesting properties regarding both of these mechanisms that could be incorporated in more comprehensive future models. First, N-803 is a latency reversing agent [41,79], which was neglected in our model. Reactivation of latent infections may have contributed to viral rebound and escape in the N-803 treated NHPs. Furthermore, the addition of PD-1 blockade, as discussed earlier, could enhance the latency reversing effect [84]. Including a latency mechanism would allow the model to more explicitly address these phenomena and inform the degree to which N-803 could reduce the latent reservoir. It would be beneficial to separately quantify the dynamics of productively infected and latently infected cells, following Banks et. al. [27]. Second, N-803 also induces cytotoxic T-cell migration into lymph tissue and B-cell follicles [15]. Our current model does not account for trafficking between blood and lymph tissue, though N-803-induced migration of CD8[+] T cells into B-cell follicles is phenomenologically represented by increases in killing rate according to parameters $\gamma_E, \gamma_K$ (Eq 7). Increased trafficking out of the blood may also have accounted for some of the observed contraction in peripheral blood CD8[+] T cells and NK cells in our NHP data. The importance of these phenomena could be more explicitly addressed by expanding the current model to include lymph node dynamics. This would allow us to ascertain how the currently predicted results from treatment improvements would translate into cloistered compartments, allowing for a better estimation of the effect of N-803 on the total body viremia.

Our model could be adapted and calibrated to data from different NHP cohorts, comparing SIV controllers and progressors or comparing N-803 responders and non-responders. While

N-803 treatment reduced the plasma SIV load in our NHPs [16], similar reductions of SIV in the plasma were not consistently demonstrated in other studies using N-803 [15] or monomeric IL-15 [9,10]. This may be because our cohort was predisposed to SIV control, which could be due to multiple factors. For example, both the Mamu-B*08 allele [85] and the Mamu-B*17 allele [86] are associated with better immune control of SIV in rhesus macaques. Beyond MHC expression, there is also evidence that CD8$^+$ T cells of human elite controllers have transcriptional signatures that favor cytokine expression over cytolytic functions, as compared to CD8$^+$ T cells from chronic progressors [87]. Mechanisms behind elite control of SIV/HIV still need to be elucidated by further experimental and modeling studies. Future mathematical models could evaluate the possible influence of MHC alleles and CD8$^+$ functionality in driving differences between these groups.

While the timing of the viral suppression and rebound with N-803 was replicated, the extent of the suppression was underrepresented in our models compared to some of the subjects' data. This may stem from some of the model assumptions or from the minimalist representation of the mechanisms in question. It should also be noted that there was significant variability in the measured response between subjects, particularly that of CD8$^+$ T cells and NK cells. Larger data sets will be needed for the models to properly characterize distributions of individual responses and make robust predictions for individuals. Nonetheless, our models were able to reproduce the varying response across different cycles of treatment that were separated by short or long timespans, which was the biological question of interest.

In summary, we developed and analyzed a mathematical model to help decode the complex immune interactions induced by N-803-therapy of HIV. This work will inform not only N-803 treatment but also its potential combination with other immune therapies and ART toward a functional cure for HIV.

## Supporting information

**S1 Fig. Sampled parameter distributions.** Panels (A-E) show the Bayesian MCMC sample of the posterior distributions of parameter values for the full model and for models #1-4 on a logarithmic scale. Bayesian 95% credible intervals are shown as dotted lines. Allowed parameter ranges (from Table 3) are shown as solid lines. Note that some units of measurement (shown below panel E) are different from those in Table 3.
(TIF)

**S2 Fig. Model comparison for cytotoxic cells (Models #2 and #4).** Panels (A,B) show fold change in CD8$^+$ T cells and NK cells in the peripheral blood, respectively, for the model without immune regulation (yellow model #2) and the model without drug tolerance or viral escape (blue model #4). The bold line corresponds to the best-fit model, and the shaded region corresponds to the Bayesian 95% credible interval. See S1 Fig for corresponding parameter distributions. Data from N-803-treated SIV-infected NHPs are shown as different symbols for each NHP [16]. Panel (C) shows timing of 0.1 mg/kg subcutaneous doses of N-803.
(TIF)

**S3 Fig. Contributions of drug tolerance and immune regulation to cytotoxic cell proliferation.** Shown are measures of mechanism contribution to CD8$^+$ T cell and NK cell proliferation for the model with immune regulation and viral escape (cyan model #1) and the model with immune regulation and drug tolerance (purple model #3). Panels (A,B) show the fold change in CD8$^+$ T cell proliferation and NK cell proliferation due to tolerance (Eq 30). Panels (C,D) show the fold change in CD8$^+$ T cell proliferation and NK cell proliferation due to regulation (Eq 29). The bold line corresponds to the best-fit model, and the shaded region corresponds to

the Bayesian 95% credible interval. See S1 Fig for corresponding parameter distributions. Panel (E) shows timing of 0.1 mg/kg subcutaneous doses of N-803.
(TIF)

**S4 Fig. Sample time courses for N-803 regimen changes for model #1.** Panels (A-C) show the results of changing the N-803 dosing frequency for the model with immune regulation and viral escape (model #1). Panel (A) shows the fold change in viral load corresponding to the 0.1 mg/kg subcutaneous dosing regimens with 2-, 3-, and 4-week dosing. Panel (B) shows the corresponding fold changes in per-cell killing due to regulation (Eq 25). Panel (C) shows the corresponding changes in the frequency of the CD8$^+$ T cell escape variant. Panels (D-F) show the response of model #1 to the 1-week dosing regimen (Fig 2) delivered along with regulatory blockade (simulated by 40, 70, and 90% reduction of killing regulation parameter λ). The bold line corresponds to the best-fit model, and the shaded region corresponds to the Bayesian 95% credible interval.
(TIF)

**S5 Fig. Sample time courses for N-803 regimen changes for model #3.** Panels (A-C) show the results of changing the N-803 dosing frequency for the model with immune regulation and drug tolerance (model #3). Panel (A) shows the fold change in viral load corresponding to the 0.1 mg/kg subcutaneous dosing regimens with 2-, 3-, and 4-week dosing. Panel (B) shows the corresponding fold changes in per-cell killing due to regulation (Eq 25). Panel (C) shows the corresponding fold changes in drug efficacy due to tolerance (Eq 24). Panels (D-F) show the response of model #3 to the 1-week dosing regimen (Fig 2) delivered along with regulatory blockade (simulated by 40, 70, and 90% reduction of killing regulation parameter λ). The bold line corresponds to the best-fit model, and the shaded region corresponds to the Bayesian 95% credible interval.
(TIF)

**S1 Appendix. Document containing additional methodological details and discussion, along with results and discussion of global sensitivity analysis and individual fitting.**
(DOCX)

**S1 Data. Spreadsheet containing model generated data used to create Figs 3–8 and S2-S5, along with experimental data used to train models and resulting parameter values (S1 Fig).**
(XLSX)

**S2 Data. Spreadsheet containing model generated data used to create figures in S1 Appendix and parameter values obtained from individual fitting.**
(XLSX)

## Acknowledgments

We thank ImmunityBio for supplying the reagent N-803.

## Author Contributions

**Conceptualization:** Jonathan W. Cody, Amy L. Ellis-Connell, Shelby L. O'Connor, Elsje Pienaar.

**Data curation:** Jonathan W. Cody, Amy L. Ellis-Connell.

**Formal analysis:** Jonathan W. Cody.

**Funding acquisition:** Shelby L. O'Connor, Elsje Pienaar.

**Investigation:** Jonathan W. Cody.

**Methodology:** Jonathan W. Cody, Amy L. Ellis-Connell, Shelby L. O'Connor, Elsje Pienaar.

**Project administration:** Elsje Pienaar.

**Resources:** Elsje Pienaar.

**Software:** Jonathan W. Cody.

**Supervision:** Elsje Pienaar.

**Validation:** Jonathan W. Cody, Elsje Pienaar.

**Visualization:** Jonathan W. Cody.

**Writing – original draft:** Jonathan W. Cody.

**Writing – review & editing:** Jonathan W. Cody, Amy L. Ellis-Connell, Shelby L. O'Connor, Elsje Pienaar.

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
