## [Decision Letter · Decision Letter 0]

13 Dec 2020

Dear Dr Pienaar,

Thank you very much for submitting your manuscript "Mathematical modeling of N-803 treatment in SIV-infected non-human primates" for consideration at PLOS Computational Biology.

As with all papers reviewed by the journal, your manuscript was reviewed by members of the editorial board and by several independent reviewers. In light of the reviews (below this email), we would like to invite the resubmission of a significantly-revised version that takes into account the reviewers' comments.

Please take particular attention to the methodological issues raised by the reviewers. These include questions of statistical correctness and also model assumptions. In addition, you should try to add to the Methods in the main paper, so that a reader can understand (even if not in detail) what you have done without needing to read all of the supplementary information. Finally, please consider the editorial guidelines about making available the critical data sets used in your study.

We cannot make any decision about publication until we have seen the revised manuscript and your response to the reviewers' comments. Particularly, how much your results may be changed due to statistical methodology improvements. Your revised manuscript is very likely to be sent to reviewers for further evaluation.

Sincerely,

Ruy M. Ribeiro

Guest Editor

PLOS Computational Biology

Rob De Boer

Deputy Editor

PLOS Computational Biology

Reviewer's Responses to Questions

**Comments to the Authors:**

Reviewer #1: In this manuscript the authors use a quantitative model to gain new insight into mechanisms by which N-803 (an IL-15 agonist) suppresses HIV infection. The model is based on ordinary differential equations that combine the pharmacokinetics and pharmacodynamics of N-803, and is calibrated using longitudinal viral, CC8+ T cell, and NK cell measurement from the peripheral blood collected in a non- human primate (NHP) study. The model is used to assess the relative contribution of three different mechanisms that are believed to contribute to reduced N-803 treatment efficacy: 1) drug tolerance (driven by the decline in IL-15 receptor expression by CD8+ T cells); 2) immune regulation (driven by a systemic anti-inflammatory response to limit immune activation); and 3) viral escape (driven by an altered amino acid sequence of targeted CD8+ T cell epitopes during N-803 treatment). Previous studies suggest all three of these mechanisms contribute to lack of efficacy of N-803, but it has been difficult to determine the relative contribution of each. Finally, the model was used to explore theoretical treatment alterations. The model suggested that compared to drug tolerance and viral escape, immune regulation was a critical mechanism that enabled viral rebound after treatment, and that treatment could be best improved with less frequent N-803 dosing and immune regulation blockade. Overall this manuscript is well written and clear, and describes a useful quantitative tool that gives new insight into mechanisms that may be most important in N-803 efficacy. It also enables testing theoretical dosing regimens, which could be valuable for prioritizing future experiments or clinical studies.

Comments:

1. It would be useful to apply additional quantitative metrics and statistical tests to support some of the key conclusions, especially those in Figure 3. One of the main conclusions of the manuscript is that immune regulation is a critical mechanism that underpins viral rebound after N-803 treatment, and this is determined from treatment response curves after different mechanisms are removed from the model (Figure 3). Rather than presenting the data qualitatively in a chart (Figure 3A), these metrics could be quantified and statistical tests could be performed to support the statement. For example, is there a quantitative metric for viral rebound that is statistically different in Figure 3C compared to the other scenarios? It does appear that some of these metrics were quantified in the appendix (Fig S5, p. 10), though a direct comparison in the main text would provide added support. This could be done similarly to support conclusions in Figures 4, 5 and 6.

2. How could antiretroviral therapy be incorporated into the model? Could this approach be be used to explore how N-803 could be effectively use without re-activating latency (McBrien at al., Science 2020). This is mentioned briefly in the discussion but could be expanded.

3. It would be helpful to provide more detail in the symbolic legend for Figure 4 (better define both cyan and magenta lines).

Reviewer #2: The authors carried out an important study to determine the contribution of an immunomodulatory IL-15 based therapy to various mechanisms affecting treatment outcomes of SIV in NHPs. The study has the potential to have high impact because it uses experimental data in combination with mechanistic mathematical models to assess biological hypotheses about the effect of the drug N-803, an IL-15 superagonist. However, as discussed in my comments below, there are several large errors in the model calibration methods that prevent the accurate interpretation of any downstream results or figures. Although all of my comments are pertinent, they are listed in order from most to least significant.

1) The AIC calculation in S7 in the supplemental is incorrectly applied in this study. The formula used by the authors is for univariate observations, but the models are being fit to multidimensional data. The authors need to use the multivariate AIC score found in "E.J. Bedrick and C.L. Tsai, Model selection for multivariate regression in small samples, Biometrics, 50 (1994), 226–231.". This multivariate AIC paper discuss applications to nonlinear regression models, which is the case here. The number of data points would be much less than 289 as stated in the supplemental material.

2) Figure 2: The shaded region represents the model output from the top 20 parameter sets, as computed by the NLL. In the supplemental it says the top 20 corresponds to the top 1% of parameter sets. This is not statistically rigorous and may be somewhat misleading to the reader as a representation of statistical uncertainty in the plots. It's not clear that the shaded regions represent a 95% credible interval. If the authors want to include a shaded region, I advise that they use a Bayesian MCMC method to estimate parameters, perhaps initializing the chain from the parameter set found with the currently reported method, and then report the 95% confidence intervals on the parameters and plot the 95% credible intervals in the figure. If confidence intervals are calculated, the authors should be sure to use the correct formulation of the variance for replicate data, which is the case here because each variable is being fit to 3 replicate data sets at the same time. See for example, "Nonlinear regression" by Seber and Wild. To ensure proper evaluation of the manuscript, the authors should be clear and write the formula they used for the NLL in the supplementary material

3) Related to comment 2, the authors should perform statistical uncertainty quantification, which they will obtain from a method such as Bayesian MCMC, in order to justify their choice of fixed vs. non-fixed parameters. The calculation of AIC scores does inform whether or not the model is overfit, but only allows the selection of the "least overfit" model. Similarly, global sensitivity analysis with PRCC does not by itself provide a measure of whether the models are overfit, and therefore whether downstream conclusions can be drawn with statistical confidence.

4) In the Model Calibration in the supplemental, the authors state "Some viral data points lay on the lower limit of detection for the viral assay. For these points, error was only counted if the model value was above the data point." The authors should provide a statistical justification for this type of heuristic with the NLL function. In my opinion, these data points should just be removed from the model calibration if the authors choose to neglect them when the model solution is below the data point anyways. This would likely not effect the parameter estimation since none of the model solutions seem to drop below these data points. Otherwise, the optimization should be treated as a censored data problem.

Reviewer #3: In "Mathematical modeling of N-803 treatment in SIV-infected non-human primates" authors investigate the impact of drug tolerance, immune regulation, and viral escape in SIV viral dynamics in-host using data from non-human primate experiments with an IL-15 superagonist, N-803.

The models presented follow CD8+ T cell and, significantly, NK cell responses in addition to make their predictions. Notably they also incorporate PK/PD of the N-803 treatment in order to gain a more accurate picture of the dynamics.

They employ their model to explore alternative N-803 dosing strategies. I especially appreciate this part -- a wide range dosing strategies are impossible to test in vivo, and validated models are the perfect way to identify candidate "best" dosing strategies.

The modeling is careful and the parameters thoughtfully derive from literature and explained in the extensive SI (though I would appreciate more detail in the main text, see end). I find the work intriguing but do have concerns outlined in the following.

The authors "calibrate" the model to data. Why not use individual fitting or some other approach, e.g. nonlinear mixed effects modeling, with the longitudinal data? Is there a clear reason why this calibration approach is to be preferred?

I note that the "calibrated" model does not explain model variability very well, e.g. the range in NK cell expansion (outlined in page 13), and it's also clear from the number of outliers in some of the data -- and that there are trends in the viral load data that are not captured by the model (e.g. Fig. 2a, S7a). Why? These discrepancies should be addressed and the use of the model in spite of these discrepancies should be justified.

I am particularly concerned by an omission the authors identify in their discussion, of latently infected cells. Activation of provirus held in a latent state of infection is thought to be *the* driver of viral rebound following treatment interruption. Authors should justify their neglect them in an investigation of post-treatment dynamics because it seems like a massive oversight. To be clear, the statement in the discussion (page 26) on it is in my opinion indequate, as it describes inclusion of latent cell dynamics as a potential improvement, and to me the choice of a model neglecting these dynamics as a valid tool to investigate post-treatment SIV viral dynamics must be defended and justified, up front.

Minor points:

In the model derivation, dependent variables X and C are not defined. One can infer that C is drug concentration from eqs 7-8 but really they should be defined.

Parameters lambda and phi and others are also not defined -- and yet they are particularly discussed? (page 11).

In the model equations (1-12) I would avoid the concentration notation. Standard models typically represent concentrations, so it's not necessary, and it's distracting.

Authors create a delay in drug tolerance via a chain of ODEs, giving an Erlang-distribution on drug tolerance in time. Could graph resulting distribution. If there were some data one could compare the distribution to, visually, would lend further support for model choices.

Authors define a per-cell killing verbally (page 16) but do not explain how it is computed in the main text. Which is frustrating as a reader, because they then rely on it to interpret results.

A detail I perhaps missed, why are CTL and NK cell populations not stimulated by foreign pathogen?

Linking together most minor comments: I would encourage revision of the methods in the main text to make it "stand alone" better. For example, "calibration" is not a well-defined process, that should be briefly outlined for the reader, who then can go to the SI for more details - otherwise the "ranges" in parameters and results comes out of nowhere. Also an indication of parameters were calibrated vs taken from literature to help with a critical reading from this study should be in the main text. Another example: I found this sentence in the SI, "Modifications to killing rate via drug stimulation, drug tolerance, and immune regulation (Eqs. 7,8) represented changes in both the frequency of cytotoxically active cells within their respective total populations and changes in the individual efficacy cytotoxically active cells." It's an important assumption for model implementation, and not at all clear from the main text.

**Have all data underlying the figures and results presented in the manuscript been provided?**

Reviewer #1: **No: **While parameter values and model details all appear to be in the supplementary appendix, there doesn't seem to be a spreadsheet with data that underlies each figure.

Reviewer #2: **No: **The data used for model calibration are not provided as a spreadsheet.

Reviewer #3: Yes

PLOS authors have the option to publish the peer review history of their article (what does this mean?). If published, this will include your full peer review and any attached files.

Reviewer #1: No

Reviewer #2: No

Reviewer #3: No
---

## [Decision Letter · Decision Letter 1]

21 Jun 2021

Dear Dr Pienaar,

We are pleased to inform you that your manuscript 'Mathematical modeling of N-803 treatment in SIV-infected non-human primates' has been provisionally accepted for publication in PLOS Computational Biology.

Best regards,

Ruy M. Ribeiro

Guest Editor

PLOS Computational Biology

Rob De Boer

Deputy Editor

PLOS Computational Biology

Reviewer's Responses to Questions

**Comments to the Authors:**

Reviewer #1: The authors have addressed all my comments and those of the other reviewers in a thorough and thoughtful manner.

Reviewer #3: The authors have addressed my concerns, I find the manuscript suitable for publication.

**Have the authors made all data and (if applicable) computational code underlying the findings in their manuscript fully available?**

Reviewer #1: Yes

Reviewer #3: Yes

PLOS authors have the option to publish the peer review history of their article (what does this mean?). If published, this will include your full peer review and any attached files.

Reviewer #1: No

Reviewer #3: No

---

## [Editor Report · Acceptance letter]

19 Jul 2021

PCOMPBIOL-D-20-01868R1 

Mathematical modeling of N-803 treatment in SIV-infected non-human primates

Dear Dr Pienaar,

I am pleased to inform you that your manuscript has been formally accepted for publication in PLOS Computational Biology. Your manuscript is now with our production department and you will be notified of the publication date in due course.

With kind regards,

Andrea Szabo
